# DRL-RNP: Deep Reinforcement Learning-Based Optimized RNP Flight Procedure Execution

**DOI:** 10.3390/s22176475

**Published:** 2022-08-28

**Authors:** Longtao Zhu, Jinlin Wang, Yi Wang, Yulong Ji, Jinchang Ren

**Affiliations:** 1School of Aeronautics and Astronautics, Sichuan University, Chengdu 610065, China; 2College of Computer Science, Sichuan University, Chengdu 610065, China; 3National Subsea Centre, Robert Gordon University, Aberdeen AB21 0BH, UK

**Keywords:** deep reinforcement learning (DRL), required navigation performance (RNP) procedure, performance-based navigation (PBN) procedure, flight control, path planning

## Abstract

The required navigation performance (RNP) procedure is one of the two basic navigation specifications for the performance-based navigation (PBN) procedure as proposed by the International Civil Aviation Organization (ICAO) through an integration of the global navigation infrastructures to improve the utilization efficiency of airspace and reduce flight delays and the dependence on ground navigation facilities. The approach stage is one of the most important and difficult stages in the whole flying. In this study, we proposed deep reinforcement learning (DRL)-based RNP procedure execution, DRL-RNP. By conducting an RNP approach procedure, the DRL algorithm was implemented, using a fixed-wing aircraft to explore a path of minimum fuel consumption with reward under windy conditions in compliance with the RNP safety specifications. The experimental results have demonstrated that the six degrees of freedom aircraft controlled by the DRL algorithm can successfully complete the RNP procedure whilst meeting the safety specifications for protection areas and obstruction clearance altitude in the whole procedure. In addition, the potential path with minimum fuel consumption can be explored effectively. Hence, the DRL method can be used not only to implement the RNP procedure with a simulated aircraft but also to help the verification and evaluation of the RNP procedure.

## 1. Introduction

As proposed by the International Civil Aviation Organization (ICAO), the performance-based navigation (PBN) procedure is a new technology for the next generation of air traffic [1]. To be specific, it refers to the aircraft’s requirements for the precision, integrity, usability, continuity, function, and other performances of the system when flying along the designated path, as per the designed instrument flying procedure or within the specified route or air space under the corresponding condition of navigation infrastructures. The PBN procedure contains two basic navigation specifications: the rules for implementation of area navigation (RNAV) and the required navigation performance (RNP) [1]. Among them, the RNAV specifies that the aircraft should be able to fly along any expected path within the coverage of the navigation signal or within the scope of data calculated by the aircraft’s avionic device or within both of the two scopes, while the RNP is the RNAV added with onboard performance monitoring and alerting (OPMA) ability. Compared with the RNAV, the RNP can realize a higher navigation precision and only depends on the global radio navigation satellite system (RNSS). Currently, one of the key technologies for flying with the RNP procedure is to have an aircraft fly along the expected path in conformance with the RNP safety specification.

The designed RNP procedure should be safe, economic, and convenient while considering the benefits to the air traffic control officers, the approach tower, the aircraft crews, the airport ridership, and the environment. A flight procedure should be put forward by the demander firstly, then undergo the coordination between the procedure designer and the procedure’s stakeholders, and iterated many times before being applied [2]. In general, a practical flight procedure is a flight path determined as per the safety specifications of the flight procedure and the constraints of the stakeholders. However, the economic optimization of this path is still worthy of study.

Deep reinforcement learning (DRL) is a technology that combines the deep learning (DL) and the reinforcement learning (RL), where the former is to perceive the environment and the latter is to solve a problem with a decision [3]. Compared with DL, the DRL mainly obtains data by interaction with the environment. This technology has been successfully applied in many aspects such as AlphaGo [4], investment [5], UAV path planning [6], and robot control [7], which is considered as one of the closest to the artificial general intelligence (AGI) approach [8,9].

With excellent performance in continuous control, the DRL method has been widely used in path planning for aircrafts or robots. Compared with the A* algorithm, particle swarm, and other traditional path planning algorithms, the DRL-based path planning algorithm has two advantages: first, it can implement the path planning without the complete information about the environment; second, it considers the dynamic or kinematic performance of the controlled object.

In this study, the DRL algorithm was proposed the first time for optimizing the flight path in the RNP procedure. To be specific, the DRL algorithm was used to have an aircraft implement the RNP procedure as per the safety specifications, based on which an optimal path with minimum fuel consumption was sought further. The main contributions of this paper can be highlighted as follows:(a)A flight controller based on multi-task deep reinforcement learning is proposed, which can solve the multi-channel coupling control problem existing in aircraft control and provide an effective solution for similar coupling control problems.(b)A DRL method named MHRL is proposed, which combines multi-task RL with hierarchical reinforcement learning (HRL) for integrated control and decision-making. MHRL can solve the problem of flight control in accordance with safety regulations and path planning considering fuel consumption.(c)The proposed work provides possible application prospects for the research that needs to consider both aircraft control and decision-making.

The remaining of this paper is organized as follows. Section 2 briefly discusses the related work, the RNP procedure, the DRL algorithm, and the Dryden wind turbulence model are introduced. In Section 3, the proposed DRL-RNP algorithm for aircraft control and path planning are presented. Section 4 describes the environment settings and structure of the model, and the simulation results and discussion are given. Finally, Section 5 provides some concluding remarks and future directions.

## 2. Background

### 2.1. Related Works

In recent years, the DRL method has been successfully applied to path planning and flight control. For example, the deep deterministic policy gradient (DDPG) algorithm in DRL was used to seek a landing strategy with the given design requirement or reward [10], the study involved the impacts of the hyper-parameters and different topologies of neural networks on landing control training of the aircraft for stably landing of an aircraft, which also pointed out the reward shaping was a challenging task, and the dynamic weight of reward may provide new prospects this study. In [11], the double deep Q-network (DDQN) algorithm was used to train an intelligent agent to control the pitching channel attitude of a fixed-wing UAV. The dynamic nonlinear attitude model of the UAV’s pitching channel and the applicable Markov decision process (MDP) were established with a certain degree of attitude control. This work pointed out how the embedded transplantation of a single hidden layer BP neural network is a potential research direction, which would be helpful for the practical application of DRL in engineering. In the literature [12], the proximal policy optimization (PPO) algorithm in DRL was used to control the nonlinear attitude of an aircraft so that the fixed-wing UAV could extend the flight envelope; this work suggested using algorithms such as SAC that can learn offline from the collected data to solve the problem of strategy transfer from simulation to the real world, which was similar to the idea of imitation learning and it was helpful to improve the basic ability of DRL agent and shorten the training time. In [13], a multi-layer RL method that enabled the autopilot in the air traffic control (ATC) simulator was proposed to guide the aircraft to the 4-D waypoint in terms of the given conditions including the latitude and longitude, altitude, heading, and arrival time, respectively. This work suggested using reward shaping to solve the problem of frequent changes in speed and heading angle. However, we think that the reward shaping itself is a complex problem, and using a lower action frequency may be more effective. In [14], the HRL method was used to form and integrate expert knowledge with the reward, in which a trained intelligent agent can conduct short-range air combat. In an artificial intelligence (AI) game held by the Defense Advanced Research Projects Agency (DARPA), the trained intelligent agent defeated the US air force aviator and ranked second. In this work, it was suggested to add a reward for task completion in the reward function to avoid the situation that the DRL agent learn to give away near victories, which we think will be helpful for the tasks that pay more attention to results.

DRL has also been intensively used in path planning. In [15], the Q-learning algorithm in the RL was used to help a robot to avoid obstacles effectively, with an optimal collision-free path planned from the starting point to the ending point planned; because this work was carried out in a simple two-dimensional experimental environment, it can only show that DRL has basic path planning ability. A DRL method for UAV path planning was introduced in [16], based on global situation information. By taking a group of situation maps as the input, this algorithm selected actions with the greedy search strategy and heuristic rules, where the performances under static and dynamic task settings were verified in the simulated experiments. This work suggested that a more realistic situation assessment model should be used in future research, which is conducive to the practical application of engineering and the implementation of UAVs tasks. To realize intelligent path planning for unmanned ships in an unknown environment, a DRL-based independent path planning model was presented in [17]. Through the continuous interaction with the environment and the use of empirical data, this model combined the DDPG algorithm with the artificial potential field (APF) to obtain improved DRL and tested it in an electronic chart platform for independent path planning with fast convergence and high stability; this work suggested considering the motion model of the ship in the future research. We think that considering vehicle’s motion model is a very important part of the path planning, which is also the reason why the JSBSim six degrees of freedom dynamic model is introduced and used in our paper. In [18], the DRL tracking of a ground target with obstacles by UAV was surveyed before proposing an improved DDPG algorithm. The simulation results revealed that it enables the UAV to effectively keep on tracing the target and avoiding obstacles. This work pointed out that the combination of rule-based method and DRL method is a future research direction, which we believe is one of the ideas to solve complex problems with DRL method. In [19], the DRL method was used to make dynamic path planning in an unknown environment; this work used lidar to obtain observations. Compared with using vision to obtain observations, lidar can be transplanted to a more complex environment without retraining network parameters. This makes us think that it may be more effective to improve the generalization performance of the path planning algorithm from the sensors than to improve the algorithm itself.

In addition, the Hamilton–Jacobi–Issacs formulations are widely used in adaptive dynamic programming gaming and path planning. According to [20,21,22], the disadvantage of Hamilton–Jacobi–Issacs formulation-based path planning is the high computational cost, especially when dealing with multi-agents, where the computational complexity will increase exponentially with the increased number of players. According to [23], for path planning problems in differential games e.g., the Pursuit–Evasion Game, Hamilton–Jacobi–Issacs formulations can produce very accurate results through theoretical proof. This can be an advantage of these methods as the neural network suffers from poor interpretability. However, we still think that this kind of method is unsuitable for the current problem as it was designed to tackle path planning as a Pursuit–Evasion Game in air combat scenarios. As a result, at least two agents are required to play different roles: chaser and evader. The problem to be tackled in this paper has only one agent for efficient path planning to lower the fuel consumption, which is not in line with the setting of the Pursuit–Evasion game, or the Hamilton–Jacobi–Issacs formulations.

Above all, the DRL algorithm performs well in both flight control and path planning. In this paper, the problem of implementing the RNP procedure and exploring a path with the minimum fuel consumption as per safety specifications is divided into two sub-problems: flight control and path planning, as detailed in Section 3.

### 2.2. JSBSim

As an open-source flight dynamics model conceived in 1996, JSBSim aims to model any aerospace vehicle without the specific program compiling and linking codes [24]. Appearing like a “black box”, this model needs to provide XML input files, including the descriptions of an aerospace vehicle, an engine, or a script. When such files are uploaded onto the JSBSim, the model can become a part of a larger simulation framework, e.g., FlightGear or OpenEaagles, to simulate the flying of the aircraft in real time, or even faster in batch processing. For each run of the JSBSim, the system can generate a data file displaying the performance and dynamics of the simulated and surveyed aircraft. Hence, the JSBSim is verified to be particularly useful in many aspects for industrial professionals as well as academia. In this study, the JSBSim was used as the flight dynamics model of an aircraft, where the A320 aircraft was used to implement the RNP approach procedure.

### 2.3. Dryden Wind Turbulence Model

In this study, the Dryden wind turbulence model [25,26], a common model used in flight-related studies, was used to simulate a more realistic approach scene with wind disturbance. This model mathematically depicts the turbulence in the model, characterizing the mathematical laws of random turbulent wind. The upwind transfer functions of this model in three directions of the coordinates of an aircraft body can be expressed as follows:(1)$$H_{u}\left(s\right)=\sigma_{u}\sqrt{\frac{2L_{u}}{\pi V}}\cdot \frac{1}{1+\frac{L_{u}}{V}s}$$
(2)$$H_{v}\left(s\right)=\sigma_{v}\sqrt{\frac{L_{v}}{\pi V}}\cdot \frac{1+\frac{\sqrt{3}L_{v}}{V}s}{{\left(1+\frac{L_{v}}{V}s\right)}^{2}}$$
(3)$$H_{w}\left(s\right)=\sigma_{w}\sqrt{\frac{L_{w}}{\pi V}}\cdot \frac{1+\frac{\sqrt{3}L_{w}}{V}s}{{\left(1+\frac{L_{w}}{V}s\right)}^{2}}$$
(4)$$L_{w}=h$$
(5)$$L_{u}=L_{v}=\frac{h}{{\left(0.177+0.000823h\right)}^{1.2}}$$
where $H_{u}\left(s\right)$, $H_{v}\left(s\right)$, and $H_{w}\left(s\right)$ represent the transfer functions of the along wind, cross wind, and vertical wind, respectively; $L_{w}$, $L_{u}$, and $L_{v}$ denote the turbulence scale lengths in the three directions; $\sigma_{u}$, $\sigma_{v}$, and $\sigma_{w}$ are the turbulence intensities in the three directions that are relevant to the aircraft’s altitude h and turbulence level, as defined below:(6)$$\sigma_{w}=0.1W_{20}$$
(7)$$\frac{\sigma_{u}}{\sigma_{w}}=\frac{\sigma_{v}}{\sigma_{w}}=\frac{1}{{\left(0.177+0.000823h\right)}^{0.4}}$$
where, $W_{20}$ represents the wind velocity at a suggested altitude of 20 ft [25]. The turbulence levels of $W_{20}$ were summarized in Table 1.

Based on the above theory, a wind turbulence model was established. The results showed that the turbulence level of an aircraft at an altitude of 500 m was light. The variations of turbulence in the three directions within 5 s of the simulation are shown in Figure 1.

### 2.4. RNP Approach Procedure

The RNP is a concrete property of the PBN to keep on a horizontal flight path, which is generally designated with a numerical value representing the precision requirement for the overall error of the system within 95% of the flight time. For example, RNP0.3 represents that the system’s overall error is up to 0.3 nautical miles within 95% of the flight time [27]. Different navigation specifications are applied in different flight stages.

In general, the RNP approach procedure is composed of four flight legs, i.e., the initial approach leg, the intermediate approach leg, the final approach leg, and the missed approach leg. The first three legs are the most important stages as shown in Figure 2.

#### 2.4.1. Basic Concepts

The initial approach leg begins at the initial approach fix (IAF) and ends at the intermediate approach fix (IF) or the final approach fix (FAF). The IAF marks the starting of the initial approach leg and the ending of the arrival leg. It is mainly used to control the descent altitude of the aircraft, and align the aircraft with the intermediate or final approach leg through certain maneuvering. In the instrument approach procedure, the initial approach has high maneuverability. An instrument approach procedure can create more than one initial approaches, but its quantity should be restricted by the air traffic flow or other navigation requirements.

The intermediate approach leg, starting from the IF and ending at the FAF, is a transition between the initial and the final approach leg. It is used for adjusting the aircraft’s appearance, velocity, and position, descending the airplane by a small altitude, and aligning it with the final approach leg to prepare for the final approach. The IF marks the ending of the initial approach leg and the starting of the intermediate approach leg.

The final approach leg is to align the airplane with the landing path, followed by descending and landing. Its instrument flight part is from the FAF to the missed approach point (MAPt). In its visual flight part, the airplane can enter the runway linearly to land, or approach the airport while circling.

#### 2.4.2. Safety Specifications

The safety specifications of the RNP approach procedure mainly involve the protection area, the obstacle clearance altitude or obstacle clearance height (OCA/OCH), the flight velocity, and the descending gradient. If an aircraft meets the safety specifications for the above factors in the whole process, it is regarded that the whole implementation procedure complies with the corresponding safety specifications.

In the RNP approach procedure, every leg has a corresponding safety protection area. The protection area (Figure 2) takes the predetermined approach path as the axis of symmetry and is divided into the main area and the sub-area. On two sides of the approach path, the main area and the sub-area occupy half of the total area respectively. Among them, the main area is delimited taking the specified flight path as the axis of symmetry. In this area, the minimum obstacle clearance (MOC) is required to increase in full. The sub-area is delimited on two sides of the main area along the specified flight path. In this area, the MOC decreases gradually.

The half width of the protection area was calculated as follows, where 1/2AW represents the half width of the protection area, XTT represents the RNP value of the corresponding leg, BV is the buffer value. The XTT values of all RNP legs, the half width of the protection area, and the BV were obtained from the literature [28].
1/2 AW = 1.5 × XTT + BV(8)

The connection between the protection areas in different legs and the division of the protection areas in the direct path leg and turning leg are detailed in the literature [28] as issued by the ICAO. The data of the protection area in the RNP approach procedure under Section 3 of this paper were also acquired from the literature [28].

The OCA/OCH in the approach leg refers to the minimum altitude determined as per relevant obstacle clearance criteria or that determined higher than the entrance elevation of the relevant runway or airport elevation. Among them, the OCA takes the mean sea level (MSL) as the baseline, while the OCH takes the entrance elevation as the baseline. The purpose of determining the OCA/OCH is to ensure that the airplane is held at a minimum safe altitude in implementing the RNP procedure to avoid collision with obstacles in the obstacle clearance area.

The MOC refers to a vertical clearance to be maintained by the airplane when flying overhead an obstacle in the safety protection area to avoid collision with the obstacle. If the protection area has no sub-area, the whole protection area should provide the full MOC. Otherwise, the main area should provide the full MOC, while the sub-area’s MOC decreases gradually from full MOC on the inner boundary to zero on the outer boundary (Figure 3). The MOC in the main area of each leg is listed in Table 2, and the MOC at any point in the sub-area of each leg was calculated as per Equation (9). Particularly, the calculated MOC of each leg was rounded to an integer, where those of the initial and intermediate approach leg were rounded up to integers at an interval of 50 m or 100 ft and that of the final approach leg was rounded up to an integer at an interval of 5 m or 10 ft.
(9)$$MOC^{\prime }=\left(L-l\right)/\left(L/2\right)\times MOC$$

In Equation (9), *L* is the width of the local protection area of this point; *l* is the vertical clearance from this point to the nominal path; MOC is the minimum obstacle clearance in the main area of this leg.

The velocities used in the legs are also specified in [28]. Among the factors with safety requirements in the RNP approach procedure, the protection area and the OCA/OCH are related to the approach procedure and the leg type, while the flight velocity and the descending gradient are relevant to the aircraft type and the leg type.

### 2.5. Deep Reinforcement Learning

#### 2.5.1. Markov Decision Process

The decision process of a DRL intelligent agent can be modeled into a Markov decision process (MDP), 〈M=S,A,P,R,γ〉, where *S*, *A*, $P\left(s_{t+1}|s_{t},a_{t}\right)$, $R\left(s_{t},a_{t}\right)$ and γ represent the state space, the action space, the state transition probability, the reward function, and the discount factor, respectively.

An RL intelligent agent sampled from the initial stage $s_{1}$ with a fixed probability of $p\left(s_{1}\right)$. It then took an action $a_{t}\in A$ at each time step t and transferred from the state $s_{t}\in S$ to the next state $s_{t+1}$, later obtained a reward $r_{t}=R\left(s_{t},a_{t}\right)$, till reaching the episode length or the termination condition under the environmental settings. The episode length at the ending time was recorded as T. The RL algorithm aimed to learn an optimum policy $\pi \left(a_{t}|s_{t}\right)$ that could maximize the agent’s expected cumulative reward $\sum_{t=1}^{T}E_{\left(s_{t},a_{t}\right)\sim \rho_{\pi }}\left[r_{t}\right]$, where $\rho_{\pi }$ is the discounted state-action visitations of π, also known as occupancies. If the state space and action space are finite, this MDP is also called a finite MDP, which was used in most of the existing studies [8,29].

#### 2.5.2. Model-Free RL

The DRL algorithm can be divided into a model-based method and a model-free method as per whether the state transition probability P and the return function R are given or not. In the model-based method, the Bayesian model approximated or over-fitted by a simple function is generally used to improve the sampling efficiency so that effective learning can be realized with a small number of samples. However, this method can hardly be applied in a scene with complex tasks and a high-dimensional state. In contrast, the model-free method requires a large amount of sampling data. However, this method can be used in a more practical scene with complex tasks and a high-dimensional state [30]. This study was conducted using the model-free DRL algorithm.

The model-free RL algorithm can further be divided into a value-based algorithm and a policy-based algorithm. In the value-based RL algorithm, the state value function was defined as $V\left(s\right)$ = $E[\underset{t}{\sum }\gamma^{t}r_{t}|s]$ to evaluate the current state (i.e., the expectation for future cumulative return in the state s). The larger the expectation is, the more favorable the current state is. The state-action value function is defined as $Q\left(s_{t},a_{t}\right)=E[\underset{t}{\sum }\gamma^{t}r_{t}|s,a]$, representing the cumulative return of taking the action a in the state s. The value function-based RL algorithm is an indirect method that obtains the optimum policy [31] (i.e., $\pi^{*}=\underset{a}{\mathrm{argmax}}\;Q\left(s_{t},a_{t}\right)$, where $\pi^{*}$ represents the optimum policy) by maximizing the $Q\left(s_{t},a_{t}\right)$. The policy-based RL algorithm is to parameterize the policy [32] and establish a policy model $\pi_{\theta }\left(a_{t}|s_{t}\right)$, wherein θ is the parameter of the policy neural network and the optimum policy can be expressed as $\pi^{*}=\underset{\pi }{\mathrm{argmax}}[\overset{T}{\underset{t=0}{\sum }}r_{t}|\pi_{\theta }\left(a_{t}|s_{t}\right)]$. Compared with the value-based algorithm, the policy-based algorithm can be used in a scene with large or continuous action space directly. However, this method faces the problem of converging to local optimum.

To overcome the shortcomings of the value-based algorithm and the policy-based algorithm, combination of these two methods were used in the Actor–Critic (AC) method [33,34]. This method is composed of an actor network and a critic network. Among them, the actor network is a policy network that receives the state input and outputs action to approximate the policy model $\pi_{\theta }\left(a_{t}|s_{t}\right)$. The critic network uses the value function to evaluate the value of the policy, i.e., input state s, output $q\left(s_{t},a_{t}\right)$, to approximate the value function $Q\left(s_{t},a_{t}\right)$. Many new algorithms are based on AC method.

## 3. The Proposed DRL-RNP Approach

In this study, the proposed DRL-RNP approach was divided into six degrees of freedom aircraft control method and path planning method (seeking the path with minimum fuel consumption under the safety specifications of the RNP approach procedure). Each RNP approach procedure leg is composed of the protection area and the expected flight path formed by connecting the waypoints. The waypoint is located at the center of the protection area and contains data on its latitude, longitude, and altitude. The width of the protection area depends on the leg type.

### 3.1. Aircraft Controller Based on Multi-Task DRL

The dynamics of a Fixed-Wing aircraft is described and simulated using JSBSim. The JSBSim flight dynamics machine (FDM) utilizes the concepts of aircraft geometry, lift, and drag forces, and the resultant moments, created by control surface actuation and environmental conditions, to simulate the motion of aircraft flight. The Equations of Motion of a Fixed-Wing aircraft is comprised of translational and rotational nonlinear equations of a 6 degree of freedom (6 DOF) system about the three axes shown in Figure 4. This generates a 12-dimensional fully observable state space, positions, and their rates, corresponding to the following state variables:(10)$$state\;variables=\left[X,Y,Z,v_{X},v_{Y},v_{Z},\phi ,\theta ,\psi ,p,q,r\right]$$

In Equation (10), X, Y, Z are linear position; $v_{X},v_{Y},v_{Z}$ are linear velocity; ϕ,θ,ψ are angular position; and p,q,r are angular velocity.

These state variables are primarily governed by the derivation of the nonlinear differential system as per [35]. The propagation of the state variables is mainly driven by inputs to the control surface deflections and throttle which are captured by the control variables.
(11)$$control\;variables=\left[\delta_{e},\delta_{a},\delta_{r},\delta_{T}\right]$$
where $\delta_{i}\in \left[-1,1\right]\;\forall \;i\in \left\{a,e,r\right\}$ is the respective normalized commanded deflections from zero position in elevator, aileron, and rudder while $\delta_{T}\;\epsilon \;\left[0,\;1\right]$ is the normalized commanded throttle command [36].

Typically, flight controllers are used to convert the control surface and throttle of aircraft into three dimensions, i.e., heading control, altitude control, and velocity control, where good results have been achieved in one or two dimensions with DRL. However, few works have considered the 3D cases. The reason is that the three dimensions have a strong coupling, which means that the control of one dimension may affect the control of the other two, making the aircraft less controllable. To tackle this challenging situation, we proposed a multi-task RL based aircraft controller. After being trained by the DRL algorithm, the proposed controller can control the heading, altitude, and velocity simultaneously and make the whole aircraft more controllable.

Although many DRL algorithms perform well in certain tasks, most of them are only specific to a certain task. Even for multiple tasks related scenes in the same environment, it was still needed to train each task separately. In this case, the data utilization efficiency was relatively low [37,38]. To solve this problem, multi-task RL has been proposed. The previous studies on multi-task RL can be divided into four categories [39]: off-policy learning of many predictions about the same stream of experience, continual learning in a sequence of tasks, distillation of task-specific experts into a single shared model, and parallel learning of multiple tasks at once. In this study, the last was used for efficiency, where we use only one neural network to achieve heading, altitude, and velocity control. We believe that for a single channel control task, even the observations of the other two channel control tasks are noisy, it can still be beneficial for the RL agent to achieve jointly learning and control of the three channels.

The flight control problem in this study can be described as shown in Figure 5. The inputs of a controller based on multi-task DRL are $id_{task}$ and state variables, in which $id_{task}$ means the expected heading or altitude or velocity, the outputs are control variables $\left[\delta_{e},\delta_{a},\delta_{r},\delta_{T}\right]$. JSBSim FDM inputs control variables and outputs state variables after dynamics model calculation. Moreover, the controller based on multi-task DRL needs to accept the reward from the reward function as input during training.

In this study, a multi-channel flight controller based on multi-task DRL was proposed. Compared with standard DRL algorithm, we add a task ID associated with the control task as input. We choose soft actor–critic (SAC) algorithm as DRL algorithm in the flight controller [40], which is the common baseline in most RL libraries.

In the multi-task SAC algorithm, the task ID is added in state:(12)$$s_{t}^{\prime }=\left(s_{t},id_{task}\right)$$

According to [40], the state value function in SAC algorithm is trained to minimize the squared residual error:(13)$$J_{V}=E_{s_{t}{}^{\prime }\sim D}[\frac{1}{2}{\left(V\left(s_{t}^{\prime }\right)-E_{a_{t}\sim \pi }[Q\left(s_{t}^{\prime },a_{t}\right)-log\pi (a_{t}|s_{t}^{\prime })]\right)}^{2}]$$
where, $V\left(s_{t}^{\prime }\right)$ is state value function, $Q\left(s_{t}^{\prime },a_{t}\right)$ is soft Q-function, $\pi (a_{t}|s_{t}^{\prime })$ is a tractable policy, and D is a replay buffer.

The gradient of Equation (13) can be estimated with an unbiased estimator:(14)$$\nabla J_{V}=\nabla V\left(s_{t}^{\prime }\right)(V\left(s_{t}^{\prime }\right)-Q\left(s_{t}^{\prime },a_{t}\right)+log\pi (a_{t}|s_{t}^{\prime }))$$
where the actions are sampled from the current policy. The soft Q-function is trained to minimize the soft Bellman residual:(15)$$J_{Q}=E_{(s_{t}^{\prime },a_{t})\sim D}[\frac{1}{2}{\left(Q\left(s_{t}^{\prime },a_{t}\right)-\hat{Q}\left(s_{t}^{\prime },a_{t}\right)\right)}^{2}]$$
(16)$$\hat{Q}\left(s_{t}^{\prime },a_{t}\right)=r_{t}+\gamma E_{\rho_{\pi }\left(s^{\prime }\right)}[V(s_{t+1}^{\prime }])]$$
(17)$$V(s_{t+1}^{\prime })=E_{a_{t}\sim \pi }[Q(s_{t+1}^{\prime },a_{t+1})-\alpha log\pi (a_{t+1}|s_{t+1}^{\prime })]$$
where α is the temperature parameter to control the stochasticity of the optimal policy, Equation (15) can be optimized with stochastic gradients:(18)$$\nabla J_{Q}=\nabla Q\left(s_{t}^{\prime },a_{t}\right)(Q\left(s_{t}^{\prime },a_{t}\right)-r_{t}-V_{target}(s_{t+1}^{\prime }))$$
the update makes use of a target value critic network $V_{target}$.

The policy network be learned by minimizing the expected KL-divergence $D_{KL}\left(\cdot \right)$ as follows:(19)$$J_{\pi }=E_{s_{t}^{\prime }\sim D}[D_{KL}(\pi (\cdot |s_{t}^{\prime })||\frac{\exp\left(Q\left(s_{t}^{\prime },\cdot \right)\right)}{Z\left(s_{t}^{\prime }\right)})]$$
where $Z\left(s_{t}^{\prime }\right)$ is the partition function that normalizes the distribution. There are several options for minimizing $J_{\pi }$. We reparametrize the policy using a neural network transformation.
(20)$$a_{t}=f\left(\in_{t};s_{t}^{\prime }\right)$$
where $\in_{t}$ is an input noise vector, sampled from some fixed distribution.

According to Equation (20), we can approximate the gradient of Equation (19) with
(21)$$\nabla J_{\pi }=\nabla log\pi \left(a_{t}|s_{t}^{\prime }\right)+\left(\nabla_{a_{t}}\log\left(a_{t}|s_{t}^{\prime }\right)-\nabla_{a_{t}}Q\left(s_{t}^{\prime },a_{t}\right)\right)\nabla f\left(\in_{t};s_{t}^{\prime }\right)$$

The Algorithm 1 exhibits a summary of the steps for applying the multi-task SAC algorithm in aircraft control:
**Algorithm 1** Multi-task SAC RL algorithm for aircraft controlInitialize the policy network parameter ϕ, value critic network parameter ψ, target                value critic network parameter ψ¯, and two soft Q-function network parameters                θ1,θ2.Initialize the replay buffer D.Initialize the task id tuple (idheading task, idaltitude task, idvelocity task).Initialize the environment.**for** each episode **do**Randomly get an idtask from (idheading task, idaltitude task, idvelocity task).        **for** each environment step **do**             Sample action at from the policy πϕat|st′*,* st′=st,idtask, obtain the next state              st+1′ and reward rt from the environment, and push the tuple st′,at,rt,st+1′ to              D.**end**      **for** each gradient step **do**Sample a batch of memories from D and update the value critic network (Equation (14)), the two soft Q-function networks (Equation (18)), the policy network (Equation (21)) and the target value network (soft-update).        **end****end**

In Algorithm 1, one actor network (policy network) and four critic networks (two value critic networks and two Q critic networks) are involved. Two Q-function critic networks are used to mitigate positive bias in the policy improvement step that is known to degrade performance of value-based methods.

The flight controller based on multi-task SAC is shown in Figure 6. Both actor network and critic network constitute of fully connected layers, including nine hidden layers for each. The adam optimizer is used for updating the networks, and associated hyper-parameters are listed in Section 4.1.2. Moreover, a squashed Gaussian policy is used to select action, which is shown in Equation (22).
(22)$${\tilde{a}}_{\phi }\left(s\right)=\tanh(\mu_{\phi }\left(s\right)+\sigma_{\phi }\left(s\right)⊙\xi ),\xi \sim N\left(0,1\right)$$

### 3.2. Path Planning Based on MHRL

In last section, a flight controller based on multi-task DRL was proposed to solve the coupling problem in multi-channel control. In this section, inspired by HRL, we proposed the multi-task HRL (MHRL) to solve the path planning problem when considering RNP flight procedure safety specifications and fuel consumption.

The HRL algorithm is effective for solving complex problems by operating at different abstraction levels of time. It is scalable and has strong capabilities in transfer learning and generalization. The HRL algorithm has been well applied in many fields such as aircraft-based air combat, automatic driving, and content recommendation.

The framework of the proposed MHRL algorithm is shown in Figure 7, which contains two modules: i.e., the top agent and flight controller based on multi-task SAC. Among them, the top agent obtains the observations about airplane states, the safety specifications of the RNP procedure, the wind turbulence, and fuel consumption, and then outputs one of the seven basic maneuvers (left turning, right turning, ascending, descending, acceleration, deceleration, and hold on), which is represented by $id_{task}$. The flight controller receives the output of the top agent and state variables from environment, then outputs the control surface command to airplane.

The network structure of top agent is the same as that in Algorithm 1, including one actor network and four critic networks. Both actor and critic network constitute of fully connected layers, including seven hidden layers for each. The hyperparameters are listed in Section 4.2.2.

Considering the complexity of the environment and tasks, iteration was not conducted between the training of the two modules in this study, namely the parameters in the flight controller module were not updated when training the top agent. The details of observations, action space, and reward function are described in Section 4.2.1.

## 4. Simulation, Experiments, and Discussion

This study mainly involved two experiments. The first was conducted on the proposed multi-task RL-based flight controller. The results have validated that the multi-task RL-based flight controller had a better control effect than the flight controller trained by single-task RL and could improve the multi-dimensional flight control effect. The second mainly revealed that the proposed MHRL method could have an aircraft to implement the RNP procedure and explore a path with minimum fuel consumption in compliance with the safety specifications of the procedure.

The simulation and experiments involved in this study were carried out on a PC device with the i7-8700K CPU, 16GB memory, and NVIDIA GeForce GTX 1070Ti.

### 4.1. Aircraft Controller Based on Multi-Task DRL

#### 4.1.1. Environment Settings

In this study, the aircraft was controlled in three dimensions: i.e., heading, altitude, and velocity. For a single-task RL controller, a new control task was set at a time interval of 150 s, which came from the default reinforcement learning control tasks of JSBSim. For heading control, a new task was designed to change ±10° from the current heading; for altitude control, a new task was designed to change ±30 ft from the current altitude; for velocity control, a new task was designed to change ±20 ft/s from the current velocity. Concerning the multi-task RL-based controller, the control task randomly selected from the above three single-tasks at a time interval of 150s. 

The above four controllers (three single-task controller and one multi-task controller) are trained on the A320 aircraft. The control surface and throttle ($\delta_{e},\delta_{a},\delta_{r},\delta_{T}$) are the action spaces. The observations of single-task controller were the difference between the currently altitude, heading, velocity and the expected altitude, heading, velocity of the aircraft ($delta_{altitude},delta_{heading},delta_{velocity}$), as well as the attitude (Euler angle), altitude, and velocity. The observations of the multi-task controller add $id_{task}$ based on the observations of single-task controller.

Regarding the setting of the reward function, it was expected to grant a large reward when the aircraft’s heading, altitude, and velocity reached the expected ranges. Moreover, a special reward was set for roll angle, with an expectation that the aircraft would not have an overlarge maneuver. The settings of the reward function are expressed as below:(23)rheading=e−∣Δheadingscaleheadingraltitude=e−|Δalttitude|scalealtitudervelocity=e−|Δvelocity|scalevelocityrroll=e−rollscalerollr=rheading*raltitude*rvelocity*rroll14
where, $r_{heading}$, $r_{altitude}$, $r_{velocity}$, and $r_{roll}$ are the rewards corresponding to the heading, altitude, velocity, and roll angle, respectively; Δheading, Δalttitude, and Δvelocity are the differences between the current values and expected values of the specific variables; roll is the roll angle; $scale_{heading}$, $scale_{altitude}$, $scale_{velocity}$, and $scale_{roll}$ are scaling indices of the specific variables to ensure that the heading, altitude, velocity, and roll angle within the allowed error range can obtain a large reward. In this study, $scale_{heading}$ was set at 3°, $scale_{altitude}$ at 10 ft, $scale_{velocity}$ at 5 ft/s, and $scale_{roll}$ at 30°. Moreover, Equation (23) is the step reward, the episode reward is calculated by Equation (24), r is step reward, k is the step number in an episode, and the initial environment settings are shown in Table 3.
(24)$$episode\;reward=\overset{k}{\underset{t=1}{\sum }}r_{t}$$

#### 4.1.2. Model Settings

The hyper-parameters in the model were set as follows: the learning rate was 0.0001 and the discount factor was 0.98; the soft update coefficient was 0.05 and replay buffer size was 20,000; the mini-batch size was 1024. This model contains nine layers of neural networks. The first two layers have 512 neurons, and the remained layers have 256 neurons. The hyper-parameters and network structures in the model are listed in Table 4.

#### 4.1.3. Simulation Results and Analysis

Figure 8 presents the reward curves of three single-task (heading, velocity, and altitude) and the multi-task over 2.0 × 10^6^ steps of simulation with the same reward function, observations, and model. The horizontal axis of Figure 8 is the step number and the vertical axis is the episode reward. Compared with using episode number as horizontal axis, using total steps as horizontal axis can better reflect the step number in an episode. As shown in Figure 8, the single-task RL controller was not available for effective control of the aircraft’s velocity or heading separately, but the altitude. The reward for multi-task RL controller was higher than single-task RL controller. Moreover, within the same step numbers, the points on the multi-task reward curve are less than the alt-task reward curve, which indicates that the step numbers in a multi-task episode is greater than that in an alt-task episode. Both episode reward and step numbers in an episode indicate that the multi-task RL algorithm is more effective than the single-task RL algorithm for the aircraft’s heading, velocity, and altitude controls.

Figure 8 is reckoned that the single-task DRL controller is not effective for the aircraft’s velocity and heading controls but the altitude control, which is consistent with the characteristics of the aircraft. For the fixed-wing aircraft, all the controls should be conducted when the aircraft is maintained at a certain flight altitude; otherwise, it is easy to cause a stall and further lose control of the aircraft. Although the reward functions of heading and velocity controls considered the altitude control (Equation (23)) (maintaining at the initial altitude), the heading or velocity change disturbed the single-task DRL controller’s perception of the altitude control so that the single-task DRL controller lost the control of the aircraft’s heading or altitude. In the first 5 × 10^5^ steps of training, the reward curve of the multi-task DRL controller was almost the same as that of the single-task (altitude control) DRL controller. Hence, it is deemed that the multi-task DRL controller learned the aircraft’s altitude control first and then its heading and velocity controls, where the later learning made the controller more robust. Consequently, the multi-task RL controller’s reward curve was higher than the single-task (altitude control) DRL controller. 

To further verify that the multi-task DRL controller is effective for the single-task DRL controller, and has high robustness, we counted the success rates of having the aircraft reach the expected heading, altitude, and velocity at different task switching frequencies with or without the multi-task DRL algorithm, as shown in Figure 9a,c,e. In the three figures, the horizontal axis represents the time interval of the heading, altitude, or velocity change; the vertical axis represents the success rate in 100 tests at this time interval. From Figure 9a,c,e we can find that with the increase in time interval of heading, altitude, and velocity change, the success rate of multi-task DRL controller can arrive about 100%, 80%, and 60%, which are better than the success rate without the multi-task DRL method controller. Although the heading and altitude control arrive a high success rate, the velocity control is not good as expected, the reason we think is that the designed velocity control task is a scene with low initial speed, in which the velocity control is more difficult. The three experiments adopted the same initial conditions as the environmental settings described in Section 4.1.1.

Further, we also observed the stability of the multi-task DRL controller in simultaneous control of the heading, velocity, and altitude. With the expectation that the controller could maintain the controls in the other two channels stable when that in one channel changed, three experiments were conducted additionally to see the changes in the other two channels when one channel changed, for example, when heading change 10°, we observed changes in altitude and velocity. In the three tests, the environment settings were the same as that described in Section 4.1.1. The simulation times were identical, 320 s; at the seconds of 80 s, 160 s, and 240 s, the controller changed the heading, altitude, or velocity +/−10°, +/−30 ft, or +/−20 ft/s from the current heading, altitude, or velocity. The results are exhibited in Figure 9b,d,f. As demonstrated, when the heading changed, the velocity fluctuated by 10 ft/s at most and the altitude by 10 ft at most; when the altitude changed, the velocity fluctuated by 10 ft/s maximally and the heading by 5° maximally; when the velocity changed, the heading fluctuated by no more than 5° and the altitude by no more than 25 ft. These results showed that the multi-task DRL controller has perfect stability. In addition, the stability of the controller also has a certain scope of application. For example, when the aircraft altitude changes a lot, the aircraft velocity cannot always remain unchanged. 

Moreover, we designed five traditional controllers (PID) to control the heading, altitude, and velocity of the aircraft, and compared them with the multi-task DRL controller. The five PID controllers are pitch, roll, heading, altitude, and velocity controller. The structures of pitch, roll, and velocity controller are shown in Figure 10, the structures of heading and altitude controller are shown in Figure 11. From the Figure 10, we can observe that the input value of the PID controller is the target pitch, roll or velocity, and the output is the control surface command ($\delta_{e},\;\delta_{a},\;\delta_{T}$). From the Figure 11, we can observe that the input of the heading or altitude PID controller is the target heading or altitude, and the output is the roll angle or pitch angle, and then the roll or pitch PID controller outputs the control surface command ($\delta_{e},\;\delta_{a}$) to realize the heading and altitude control.

The results of the pitch and roll PID controller are shown in Figure 12. Figure 12a is pitch control and Figure 12b is roll control. Figure 13 compared the PID controller and multi-task DRL controller. From Figure 13, we can find that for heading control, the rise time, adjustment time, and overshoot of the multitask DRL controller are better than those of the PID controller, but the fluctuation is large after stabilization. For altitude and velocity control, the overshoot and stability of multi-task DRL controller are obviously better than PID controller. It is worth noting that coupling often occurs in the parameter adjustment of PID controllers, that is, the parameter adjustment of one PID controller affects the other PID controller. The results of the PID controller are shown in Figure 12 and Figure 13 have undergone about 80 parameter adjustments, and the final parameters are shown in Table 5. In general, we think that the multi-task DRL controller is superior to the PID controller in terms of stability and has more advantages in dealing with control coupling problems.

### 4.2. Path Planning Based on MHRL

In this part, two experiments were conducted: the first was mainly to test whether the MHRL method could meet the safety specifications or not when implementing the RNP approach procedure; the second was mainly to test the MHRL method’s exploration of the path with minimum fuel consumption in compliance with the safety specifications of the RNP approach procedure.

#### 4.2.1. Environmental Settings

In these two experiments, it was expected that the MHRL method could complete the initial and intermediate approach stages of the RNP procedure. According to the literature [27], two routes were designed, similar to a half of a “Y” shaped approach procedure (Figure 14). The longitude, latitude, half width, altitude, and velocity limits of the waypoints were listed in Table 6, where $\underline{\bar{5000}}$ means that the altitude of the aircraft at this waypoint is about 5000 ft, $\underline{2300}$ means that the altitude of the aircraft at this waypoint is not less than 2300 ft. The initial settings of the aircraft in Table 7, where the initial position of the aircraft was at the first waypoint of the approach routes, and the heading was the direction of the initial approach leg.

The observations of top agent can be divided into four parts: RNP flight procedure safety specification, wind turbulence, fuel consumption, and airplane state. RNP flight procedure safety specification contains three-part limits: altitude limit, velocity limit, and distance to expected path limit. Moreover, the leg type and distance to next waypoint are also be observed. Wind turbulence contains along wind speed, cross wind speed and vertical wind speed. Fuel consumption is the remaining fuel at every simulation step. Airplane state contains altitude and roll angle, which is less than the airplane state variable observations in flight controller agent. The observations are as follows:$$obs=\left[\Delta alt_{max},\Delta alt_{min},\Delta v_{max},\Delta v_{min},\Delta dis_{path},\Delta dis_{nextwpt},v_{along-wind},v_{cross-wind},v_{vertical-wind},fuel,alt,roll\right]$$
where $\Delta alt_{\max\left(min\right)}$ is the difference between airplane altitude and the leg max (min) altitude limit, $\Delta v_{\max\left(min\right)}$ is the difference between airspeed and the leg max (min) velocity limit, $\Delta dis_{\mathrm{path}}$ is the distance to expected path, $\Delta dis_{\mathrm{nextwpt}}$ is the distance to next waypoint, $v_{\mathrm{along}\left(\mathrm{corss},\mathrm{vertical}\right)-\mathrm{wind}}$ is the along (cross, vertical) wind speed, fuel is the remaining fuel, alt is airplane altitude, roll is roll angle. In addition, the observations are normalized in the implementation.

The action space of top agent contains seven $id_{task}$, which mean seven basic maneuvers: left turning, right turning, ascending, descending, acceleration, deceleration, and hold on. These seven basic maneuvers are combined by heading, altitude, and velocity control. We set the value of left (right) turning is 10°, the value of ascending (descending) is 30 ft, the value of acceleration (deceleration) is 20 ft/s, which are the same as the control task of flight controller in Section 4.1.1. The action space as follows:$$action\;space=\left[id_{left\;turning},\;id_{right\;turning},id_{ascending},id_{descending},id_{acceleration},id_{deceleration},id_{hold\;on}\right]$$

The reward function of top agent is divided into two parts: the first part is the reward for complying the flight procedure under safety specifications, which is the main part, the second part is the reward for the remained fuel mass. The settings of the reward function are expressed as below:(25)rAW/2=1,ifΔdispath<12AW,else1×10−4raltitude=1,ifaltmin<altairplane<altmax,else1×10−4rvelocity=1,ifvmin<vairplane<vmax,else1×10−4rdis2nextwpt=e−Δdisnextwptscaledisrfuel=e−Δfuelscalefuelr1=rAW2*raltitude*rvelocity*rdis2nextwpt14r2=r1+rfuel
where rAW2, $r_{altitude}$, $r_{velocity}$, and $r_{dis2nextwpt}$ are the first part reward, $\frac{1}{2}AW$ is the protection zero width, which is shown in Figure 2. and calculated by Equation (8), $alt_{\min\left(max\right)}$ is the leg min (max) altitude limit, $v_{\min\left(max\right)}$ is the leg min (max) velocity limit, the $scale_{dis}$ we set 5000 m, which is about half distance of two waypoints. $r_{fuel}$ is the second part reward, Δfuel is the current fuel consumption, $scale_{fuel}$ is the scale index, for route 1, we set it 252 lbs, for route 2, we set it 298 lbs, which are the fuel consumption without consideration of fuel consumption reward. $r_{1}$ is the step reward without consideration of fuel consumption, $r_{2}$ is the step reward with consideration of fuel consumption.

#### 4.2.2. Model Settings

The RL algorithm of the top agent also adopted the SAC algorithm. Compared with the SAC algorithm model in the multi-task RL controller, this model had fewer layers of networks. Its hyper parameters were set as follows: the learning rate was 0.0001 and the discount factor was 0.98; the soft update coefficient was 0.05 and the replay buffer size was 20,000; the mini-batch size was 1024. This model contained seven layers of neural networks. The first two layers had 512 neurons and the remained layers had 256 neurons. The hyper-parameters and network structures of the model are provided in Table 8.

#### 4.2.3. Simulation Results and Analysis

The reward curves for implementing the RNP procedures of route 1 and route 2 with and without MHRL method are illustrated in Figure 15. The implementation of without MHRL is to add the rewards of safety specification and fuel consumption to the reward function in the multi-task DRL controller. As shown in Figure 15, the rewards for implementing route 1 and route 2 with MHRL method were both significantly higher than those without MHRL. At the initial time of the training, the rewards for implementing the procedure with MHRL method were higher than that without MHRL method, the reason of this is that we think the top policy in MHRL method could output action to keep the aircraft stable and obtain a reward even though the top agent had not learned the effective policy.

Figure 16 shows the details of implementing the initial and intermediate approach procedures of route 1 and route 2 with MHRL method. In the following experiment, the Dryden wind turbulence model was used to simulate the impact of wind on the aircraft. The wind turbulence had three levels: light, moderate, and severe. The wind turbulence velocity at each level is calculated according to the altitude and airspeed of the aircraft, Equations (1)–(7) and Table 1. Refs. [25,26] describe in detail the equation derivation and code implementation of Dryden wind turbulence model.

For route 1, the MHRL method could have the aircraft fly along the expected path with the three degrees of wind turbulence. In the whole process, the aircraft’s altitude, velocity, and distance to the expected path complied with the safety specifications of the RNP procedure. For route 2, the aircraft was also maintained flying along the expected path with MHRL method at the three degrees of wind turbulence. In the whole process, the aircraft’s altitude and the distance to the expected path satisfied the safety specification of the RNP procedure. Although an overspeed event occurred due to the rapid decline of aircraft altitude in the first 50 s, the top agent can reduce the speed by climbing altitude to meet the safety specifications. In addition, in the second half of route 2, the top agent seems to learn to slowly reduce the height and speed at the same time.

It also can be seen from the Figure 16 that the influence of wind turbulence on the distance to expected path is more obvious than that of wind turbulence on velocity and altitude. The reason is that we think it is related to the optional actions of top agent. Turning left and right cannot well offset the impact of crosswind on aircraft flight. Flying in a sideslip attitude may achieve better results, which is a direction that this research can improve in the future.

The above results are based on without consideration of fuel consumption ($r_{1}$ in Equation (25)), then the MHRL method was taken to explore a path with minimum fuel consumption ($r_{2}$ in Equation (25)) in implementing the flight procedure. The experimental results of route 1 and route 2 are provided in Figure 17 and Table 9, Figure 18 and Table 10.

Figure 17 exhibits the flight paths of route 1 obtained by MHRL method with and without fuel consumption reward under windless conditions. The color bar shown on the right side represents the flight altitude. As observed, the two paths were less different in the initial approach stage of route 1 and largely different in the intermediate approach stage; moreover, the top agent in the MHRL method attempted to reduce fuel consumption by shorter flight path. Table 9 illustrates the fuel consumptions of MHRL method with and without fuel consumption reward at four wind levels: the first row presents the fuel consumption without fuel consumption reward, and the second row presents the fuel consumption with fuel consumption reward; the fuel consumptions shown in the table are the averages of 50 tests obtained at four wind levels; the last row presents the percentage of reduction in the fuel consumption (the fuel consumption difference between the two cases divided by the fuel consumption without fuel consumption reward). It can be seen that the MHRL method with fuel consumption reward could help reduce about 5% fuel consumption for route 1.

Figure 18 displays the flight paths of route 2 obtained by MHRL method with and without fuel consumption reward under windless conditions, where the right color bar represents the flight altitude. As observed, the two paths were largely different in the initial approach stage of route 2 and less different in the intermediate approach stage; compared with the path without fuel consumption reward, the path with fuel consumption reward can reduce fuel consumption by selecting appropriate turning time and further reducing the flight path. Table 10 lists the fuel consumptions of MHRL with and without fuel consumption reward at four wind levels: the first row presents the fuel consumption without fuel consumption reward, and the second row presents the fuel consumption with fuel consumption reward; the fuel consumptions shown in the table are the averages of 50 tests obtained at four wind levels; the last row presents the percentage of reduction in the fuel consumption (the fuel-consumption difference between the two cases divided by the fuel consumption without fuel consumption reward). It can be observed that the MHRL method with fuel consumption reward contributed to reducing about 13% fuel consumption for route 2.

It can be seen from Table 9 and Table 10 that whether for route 1 or route 2, the higher the wind turbulence level, the more fuel consumption. The reason for this is, we believe, related to the frequent change of maneuvers and the increase in flight path under wind turbulence.

In addition, another topic worth discussing is the design of reward function considering fuel consumption ($r_{2}$ in Equation (25)), which is also an important and difficult part in DRL. In fact, we tried several different reward functions to reduce fuel consumption, for example, taking $r_{fuel}$ (in Equation (25)) as an item in $r_{1}$ (in Equation (25)), only taking $r_{fuel}$ as $r_{2}$ (in Equation (25)), etc.; however, none of them obtain better results than $r_{2}$, and some even lead to worse results than ignoring fuel consumption. This makes us have to consider that this only is the lowest fuel consumption path under $r_{2}$, not for route 1 or route 2.

In summary, the proposed MHRL method can control aircraft execute the RNP approach procedure while satisfying the safety specification. Meanwhile, the MHRL method with fuel consumption reward is effective for exploring a path with minimum fuel consumption, and the basic maneuver of top agent in MHRL method and the reward function with fuel consumption are the improvement directions of this study.

## 5. Conclusions

In this study, the multi-task extended HRL (MHRL) algorithm was proposed for implementing the RNP procedure and exploring a path with minimum fuel consumption (i.e., for solving the problems of aircraft control and path planning). To solve the multi-channel coupling control problem in aircraft control, a flight controller based on multi-task DRL was raised to learn the generalities of the three dimensions of control (heading, altitude, and velocity). As for the path planning for minimizing the fuel consumption, the MHRL algorithm was suggested. In this algorithm, the bottom layer is for the aircraft control, while the top agent is for exploring a path with minimum fuel consumption. The simulation experimental results revealed that the multi-task DRL controller can realize end-to-end control of an aircraft and can be used to solve the coupling control problem in similar scenes. The MHRL algorithm can control aircraft to explore a path with the minimized fuel consumption while complying with the safety specifications, offering an idea for RNP procedure design and verification.

As also shown, the MHRL algorithm can solve a complex problem effectively by having the problem divided into several sub-problems and conquering them separately. It is hopeful to apply this method to solve complex problems in the aviation field. Nevertheless, the proposed algorithms and experiments also have some limitations. In addition to the altitude and velocity, more restrictions (e.g., the descending rate and the turning rate) should be considered in implementing the RNP procedure. Moreover, the exploration of a path with minimum fuel consumption was conducted on the given RNP procedure. In the practical design of the RNP procedure, the departure and destination points are the considerations for judging the minimum fuel consumption. It is expected to solve these problems by setting a more reasonable reward function in combination with traditional path planning algorithms, and comparing the method proposed in this paper with the lowest fuel consumption path obtained from historical flight data is the future work of this paper, the acquisition of more real and reliable historical flight data is a difficult problem, using data from a simulated air traffic control system maybe is a solution.

## Figures and Tables

**Figure 1 sensors-22-06475-f001:**
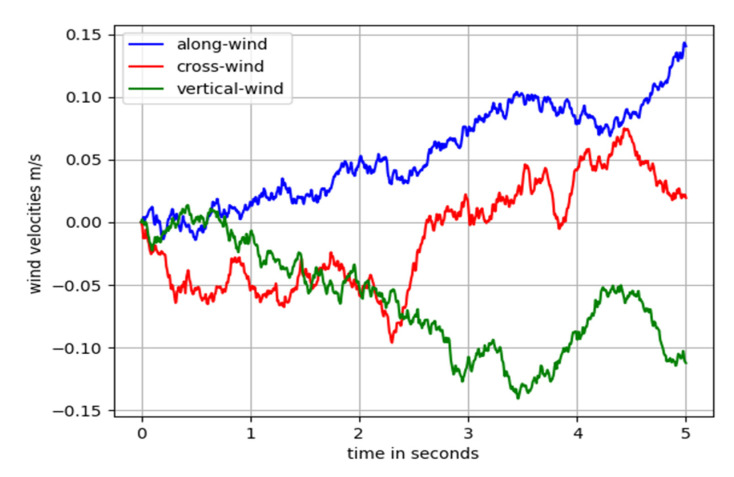
Wind turbulence profile.

**Figure 2 sensors-22-06475-f002:**
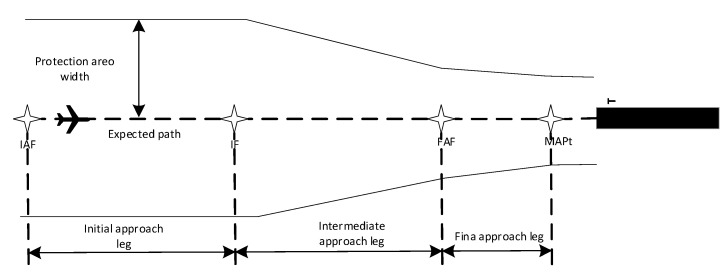
Legs of the RNP approach procedure.

**Figure 3 sensors-22-06475-f003:**
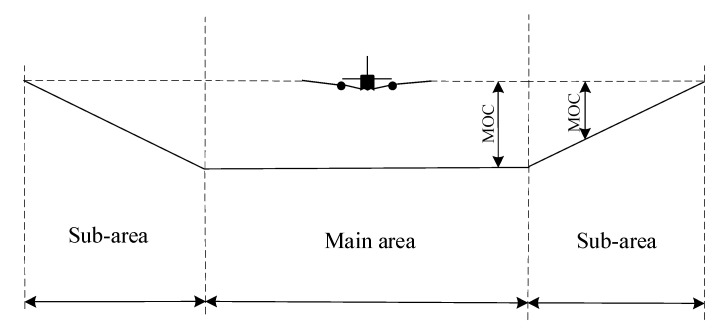
Protection area interface main area and sub-area and corresponding MOC.

**Figure 4 sensors-22-06475-f004:**
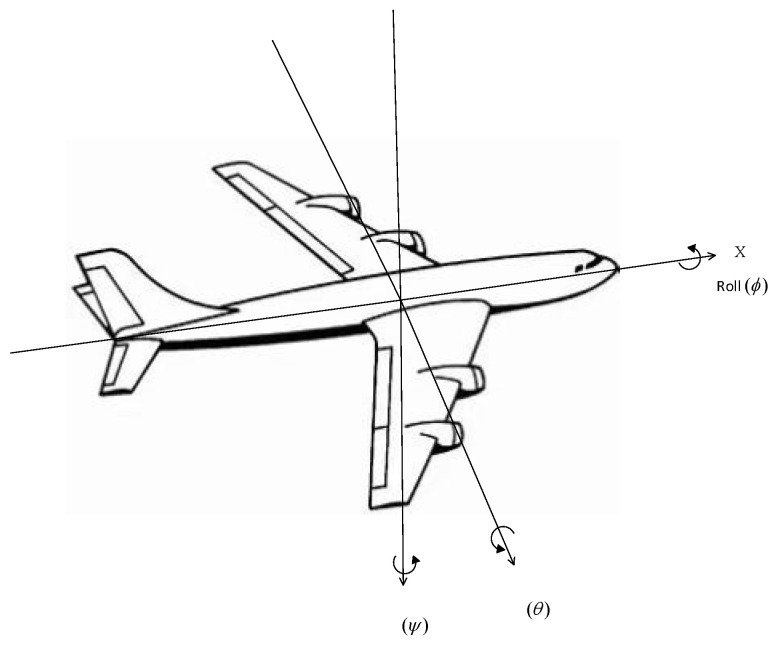
Diagram of body axes denoting the states of an aircraft.

**Figure 5 sensors-22-06475-f005:**
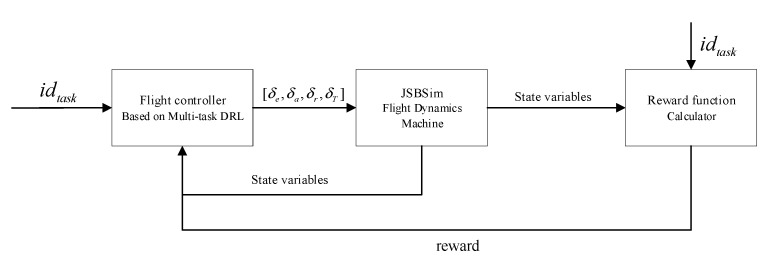
Overview of flight control based on DRL.

**Figure 6 sensors-22-06475-f006:**
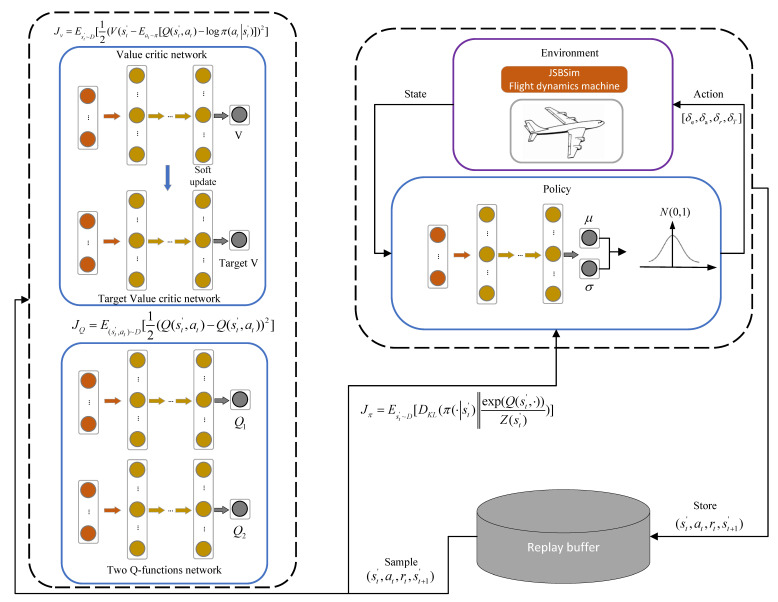
The flight controller based on multi-task SAC.

**Figure 7 sensors-22-06475-f007:**
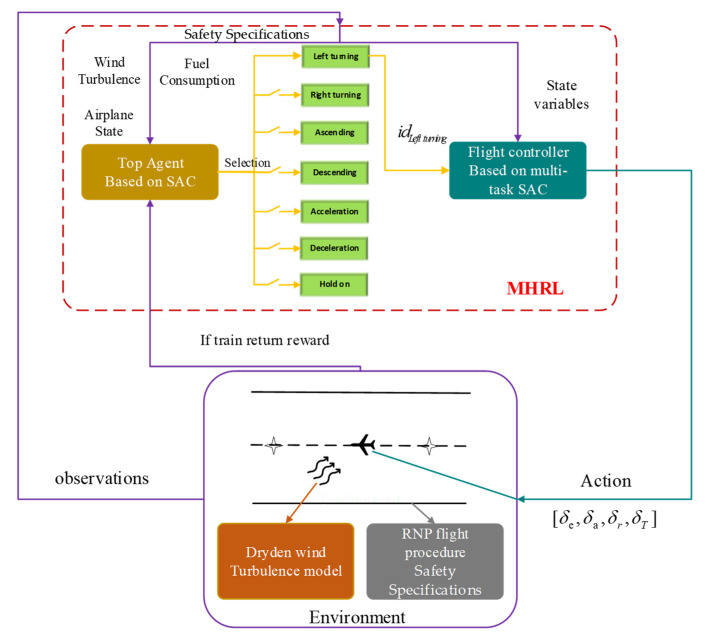
Framework of the MHRL algorithm.

**Figure 8 sensors-22-06475-f008:**
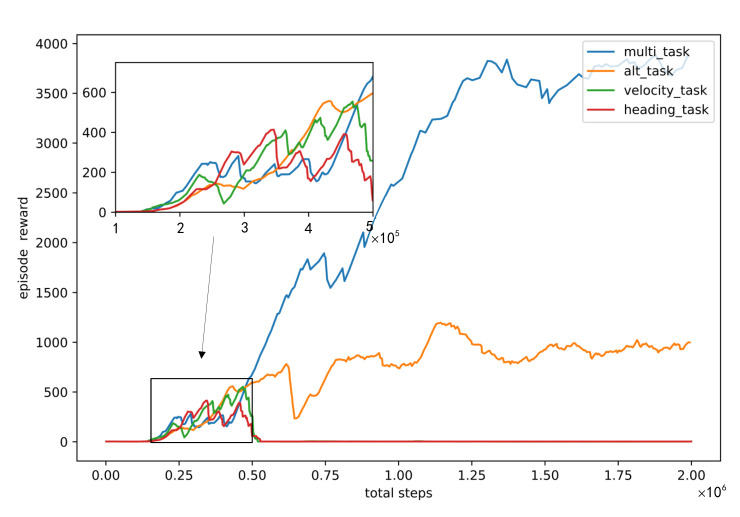
Single-task and multi-task training reward curves. After 2 × 10^6^ steps training, the reward of multi-task DRL controller can be close to 4000, the reward of altitude-task DRL controller is about 1k, the rewards of velocity-task and heading-task are close to 0.

**Figure 9 sensors-22-06475-f009:**
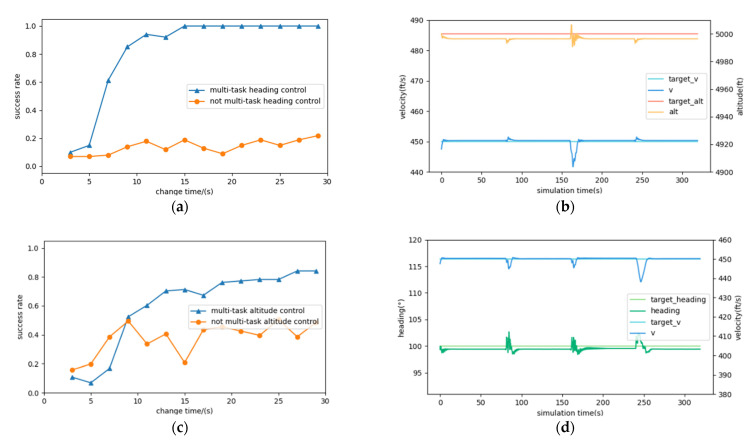
The success rates in the 100 tests on the heading, altitude and velocity controls with multi-task RL algorithm (**a**,**c**,**e**) and the controller’s control in the other two dimensions when the heading, altitude, or velocity changes (**b**,**d**,**f**).

**Figure 10 sensors-22-06475-f010:**
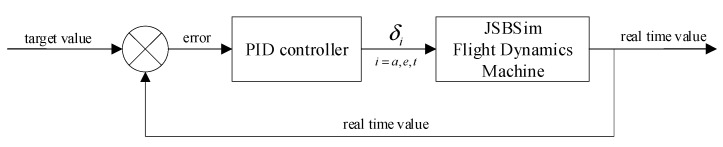
The structures of pitch, roll, and velocity controller.

**Figure 11 sensors-22-06475-f011:**
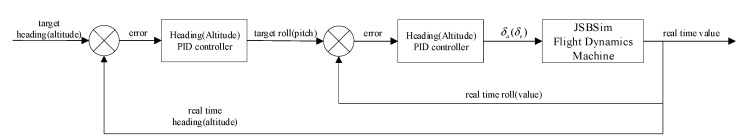
The structures of heading and altitude controller.

**Figure 12 sensors-22-06475-f012:**
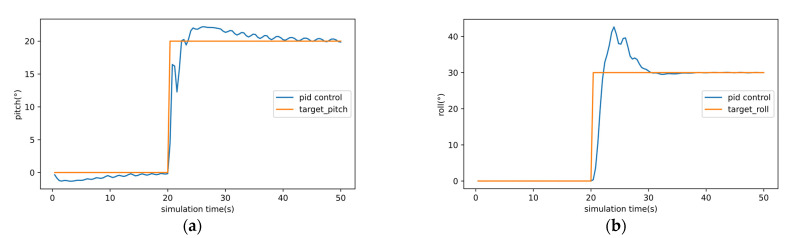
The results of the pitch and roll PID controller. (**a**) is the pitch angle tracking, (**b**) is the roll angle tracking.

**Figure 13 sensors-22-06475-f013:**
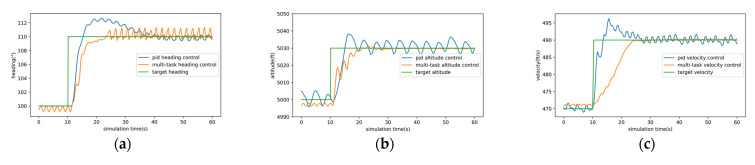
The comparison PID control and multi-task control. (**a**) is the comparison of heading control, (**b**) is the comparison of altitude control, (**c**) is the comparison of velocity control.

**Figure 14 sensors-22-06475-f014:**
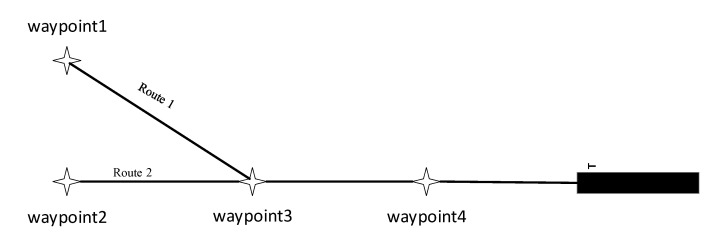
The approach routes.

**Figure 15 sensors-22-06475-f015:**
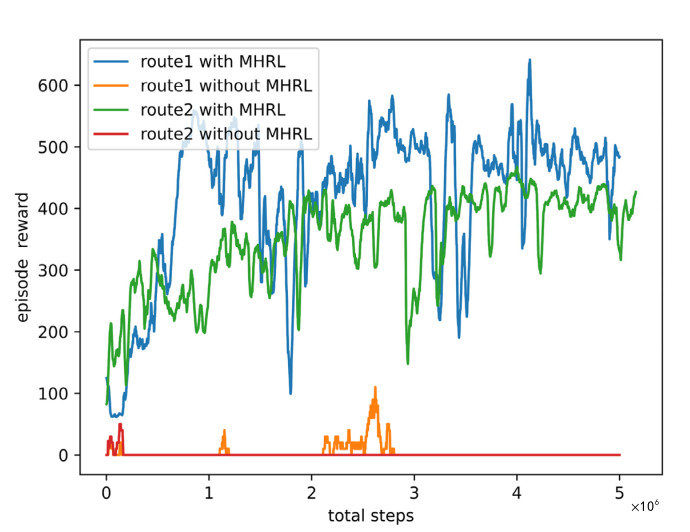
The curves of rewards for implementing routes 1 and 2 with and without MHRL method. After 5e6 steps training, for route 1, the reward can reach about 500 with MHRL method, the reward is about 0 without MHRL method. For route 2, the reward can reach about 400 with MHRL method, the reward is about 0 without MHRL method.

**Figure 16 sensors-22-06475-f016:**
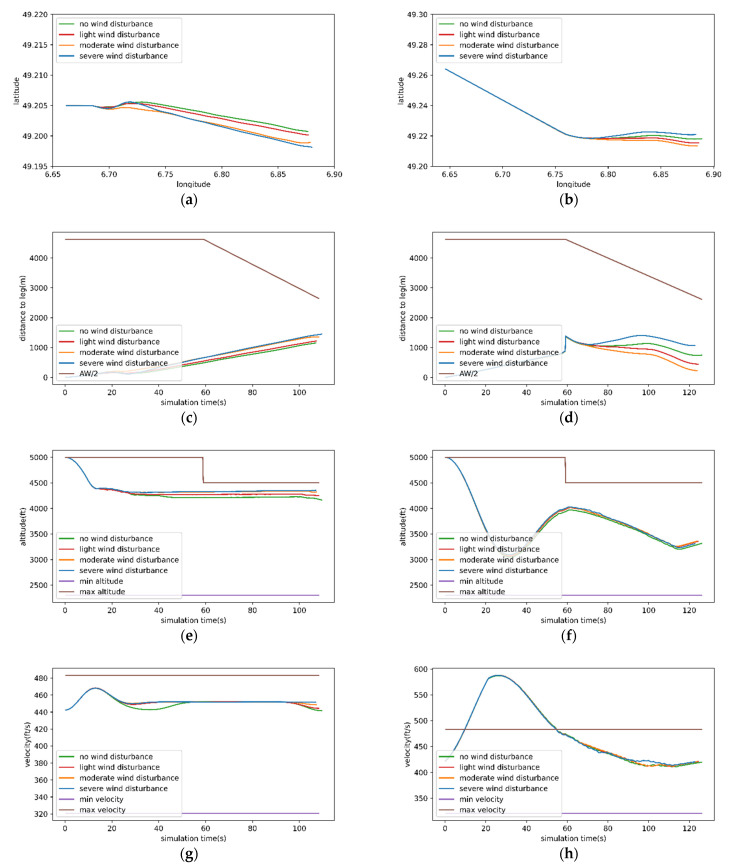
Details in the implementation of route 1 and route 2 with the MHRL method: (**a**) is the flight paths of implementing route 1 with MHRL method under four wind turbulence levels; (**c**,**e**,**g**), are the varieties of the distance to expected path, altitude, and velocity corresponding to (**a**); (**b**) the is flight paths of implementing route 2 with MHRL method under four turbulence wind levels; (**d**,**f**,**h**) the varieties of the distance to expected path, altitude, and velocity corresponding to (**b**).

**Figure 17 sensors-22-06475-f017:**
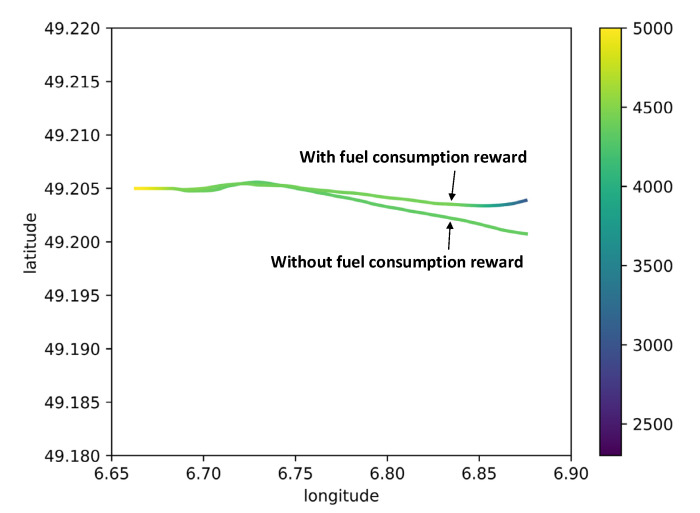
Route 1 flight path with and without fuel consumption under no wind conditions, the color bar shown on the right side represents the flight altitude.

**Figure 18 sensors-22-06475-f018:**
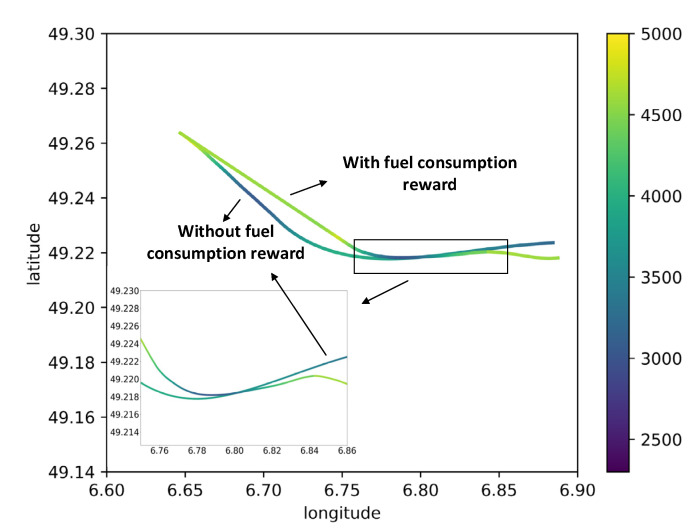
Route 2 flight path with and without fuel consumption under no wind conditions, the color bar shown on the right side represents the flight altitude.

**Table 1 sensors-22-06475-t001:** Wind velocity of the wind levels.

Turbulence Level	$W_{20}$
Light Moderate Severe	15 knots 30 knots 45 knots

**Table 2 sensors-22-06475-t002:** MOC in the main area and sub-area.

Leg Type	Main Area	Sub-Area
Initial approach leg Intermediate approach leg Final approach leg	300 m 150 m 75 m	0–300 m 0–150 m 0–75 m

**Table 3 sensors-22-06475-t003:** The Initial Settings of the Environment.

Parameters	Value
Aircraft type Initial latitude Initial longitude Initial altitude Terrain altitude Initial velocity Initial heading Initial roll angle $\delta_{T}$ $\delta_{a}$ $\delta_{e}$ $\delta_{r}$	A320 49.392057° N 7.057191° E 5000 ft 1022 ft 450 ft/s 100° 0° 0.8 0 0 0

**Table 4 sensors-22-06475-t004:** Parameter Settings of the Model.

Parameters	Value
Learning rate Discount factor Soft update coefficient Replay buffer size Minibatch size Network frame Activation function	1 × 10^−4^ 0.98 0.05 20,000 1024 [256,256,256,256,256,256,256,512,512] Relu

**Table 5 sensors-22-06475-t005:** The parameters of five PID controller.

The Types of PID Controller	P	I	D
Roll controller Pitch controller Velocity controller Heading controller Altitude controller	1.3 2.9 2.1 2.7 2.7	0.19 0.13 0.15 0.17 0	1.9 0.3 5.0 1.1 3

**Table 6 sensors-22-06475-t006:** Data of the Waypoints.

Point	Latitude	Longitude	Altitude	AW/2	Velocity
Waypoint 1 Waypoint 2 Waypoint 3 Waypoint 4	N 49.26417 N 49.205 N 49.20806 N 49.21167	E 6.64556 E 6.66167 E 6.76889 E 6.89778	$\underline{\bar{5000}}$ ft $\underline{\bar{5000}}$ ft $\underline{2300}$ ft $\underline{2300}$ ft	4360 m 4360 m 4360 m 2685 m	320<V<483 ft/s 320<V<483 ft/s 320<V<483 ft/s 320<V<483 ft/s

**Table 7 sensors-22-06475-t007:** Initial States of the Aircraft.

Parameters	Value
Aircraft type Initial position Initial altitude Initial velocity Initial heading Initial roll angle Initial fuel weight	A320 Waypoint1(2) 5000 ft 450 ft/s 90° (120°) 0° 15,000 lbs

**Table 8 sensors-22-06475-t008:** The Hyper-Parameters of the MHRL Method’s Top Agent.

Parameters	Value
Learning rate Discount factor Soft update coefficient Replay buffer size Minibatch size Network frame Activation function	1 × 10^−4^ 0.98 0.05 20,000 1024 [512,512,256,256,256,256,256] Relu

**Table 9 sensors-22-06475-t009:** Route 1 average of 50 tests of fuel consumption under four wind levels.

	No Wind	Light Wind	Moderate Wind	Severe Wind
Without fuel consumption reward	252.16 lbs	256.78 lbs	258.19 lbs	262.46 lbs
With fuel consumption reward	237.76 lbs	235.67 lbs	244.24 lbs	247.13 lbs
Difference	−5.71%	−8.22%	−5.40%	−5.84%

**Table 10 sensors-22-06475-t010:** Route 2 average of 50 tests of fuel consumption under four wind levels.

	No Wind	Light Wind	Moderate Wind	Severe Wind
Without fuel consumption reward	298.30 lbs	298.06 lbs	296.61 lbs	300.56 lbs
With fuel consumption reward	256.78 lbs	257.52 lbs	260.32 lbs	263.73 lbs
Differences	−13.91%	−13.06%	−12.23%	−12.25%

## Data Availability

Not applicable.

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
