# Peer review of "DRL-RNP: Deep Reinforcement Learning-Based Optimized RNP Flight Procedure Execution"

_sensors, 2022, doi:10.3390/s22176475_

Round 1
Reviewer 1 Report
- The problem is not defined clearly. In other words, since RL is a decision-making method, it is important to define what the control variables are and what the output (or state variables) is. The authors have to define the problem such that the specified output becomes the desired one by designing the specified control variables. Currently, although there are so many big words like hierarchical RL and multi-task RL in the paper, in fact, the control problem under consideration is not defined clearly.
- This is also about the description of the problem. In section 2.3, the authors presented many things about wind turbulence, for example, equations (1)-(7). However, the equations are not referred to later. If this is true, why the authors wrote the equations. Those unnecessary details can make the flow of the paper messy.
- The main result of the paper is deep RL. To explain the ‘deep’ part, the authors have to present the details of the neural networks. Since SAC is used in the paper and SAC consists of 5 neural networks, there must be details of 5 neural networks but there is only one in the paper. This makes me doubt if SAC is properly applied in the paper. In addition, to present RL, the most important part is to define MDP clearly for the problem. Namely, what the state, action, and reward functions are for the problem. There are some but the authors have to define the elements of MDP for the problem more clearly and explicitly. Moreover, it is necessary to depict the feedback loop describing how the SAC-based agent interacts with the environment in the problem since the problem (RNP) itself looks pretty complicated.
- The reward function looks a bit confusing. The authors need to explain more the rationale behind the definition of the reward function in (16). In (16), for example, if delta heading is large (i.e. the error is large), then the corresponding reward r_{heading}=exp(-delta heading/scale) becomes small, and if delta heading is very small, r_{heading}=exp(-delta heading/scale) becomes almost 1. If this is right, if all variables are well controlled after long training, all delta variables become almost zero. As a result, all reward functions become 1. Extremely speaking, after long training, the agent is well trained, the deviation variables (delta ..) become 0 and the corresponding reward functions become 1. Then how the reward curves in Figure 5 can be generated. Or simply, the range of the reward functions is (0,1). Then, how the curves in Figure 5 can be made?
- The authors used RNP (i.e. abbreviation) in the title. I am not sure how many people can see the definition of RNP.
Reviewer 2 Report
The subject treated by the authors is very interesting and the presented results are encouraging, but the numerous writing errors make it difficult to understand the proposal and originality of the work. In addition, I would like to make following comments:
1. The authors need to improve the literature review, describing completely the proposal of each work and, after that, to point out the novelty of their proposal.
2. The authors need to make very clear the proposed method. An overall review in the text is necessary to describe all symbols and acronyms used.
3. The discussions of the results are very poor. The authors need to expand and better describe the experiments.
4. Finally, This paper discusses some optimization algorithms but doesn't seem to have much to do with sensors?...
Reviewer 3 Report
The paper presents an interesting, deep RL based approach to path planning, which is an important topic that has practical applications in air traffic control systems. However, the literature review on path planning seems far from adequate. While RL is a useful methodology, other schools of thought have been studied and have proven useful under non-stationary obstacles setting. In particular, Hamilton-Jacobi-Issacs formulations of path planning provide an important class of methods, which the authors should thoroughly discuss in the introduction and position their work in:
1) Efficient path planning algorithms in reach-avoid problems, Zhou et al, Automatica, 2018
The authors need to discuss the merits and weaknesses of this class of methods and why it would or would not apply to their current problem.
Round 2
Reviewer 1 Report
The authors answered all my questions.
One suggestion is that, for comparison purposes, the authors had better describe how the conventional control methods deal with the problem under consideration. It might be very good to contain qualitative/quantitative comparisons with the results by the conventional control method.
Reviewer 2 Report
no comments for this version
Author Response
Thanks your comments :).
Reviewer 3 Report
the authors didn't seem to address my comments
Round 3
Reviewer 1 Report
The authors answered all my comments. The paper was ready for publication.
Reviewer 3 Report
My concerns are addressed. Recommend accept.